# Human Metapneumovirus: A Narrative Review on Emerging Strategies for Prevention and Treatment

**DOI:** 10.3390/v17081140

**Published:** 2025-08-20

**Authors:** Nicola Principi, Valentina Fainardi, Susanna Esposito

**Affiliations:** 1Università degli Studi di Milano, 20122 Milano, Italy; nicola.principi@unimi.it; 2Pediatric Clinic, Pietro Barilla Children’s Hospital, Department of Medicine and Surgery, University Hospital of Parma, 43126 Parma, Italy; valentina.fainardi@unipr.it

**Keywords:** human metapneumovirus, respiratory infections, monoclonal antibodies, vaccine development, antiviral therapy, pediatric respiratory disease

## Abstract

Human metapneumovirus (HMPV) is a major cause of acute respiratory tract infections, particularly in infants, young children, older adults, and immunocompromised individuals. Since its discovery in 2001, the virus has been recognized for its significant clinical and socioeconomic impact. Despite extensive research, no licensed vaccines or antiviral therapies are currently available for HMPV. This review aims to synthesize current knowledge on HMPV prevention and treatment, and to highlight promising avenues for future interventions. Several monoclonal antibodies (mAbs) targeting conserved epitopes of the HMPV fusion (F) protein have shown strong neutralizing activity in vitro and in animal models, although none have reached clinical trials. Vaccine development, including subunit, live attenuated, vector-based, and mRNA platforms, is progressing, with some candidates showing promise in adult populations. However, data in children, especially seronegative infants, remain limited. Antiviral research has explored repurposed drugs such as ribavirin and probenecid, along with novel agents like fusion inhibitors and T-cell-based immunotherapies, though none are yet approved. The development of safe, effective interventions—especially multivalent approaches targeting multiple respiratory viruses—remains a high priority. Continued research is essential to bridge the gap between preclinical promise and clinical application and to reduce the burden of HMPV infection worldwide.

## 1. Introduction

Since its discovery in 2001 by van den Hoogen et al. [1], human metapneumovirus (HMPV) has been recognized as a significant cause of both upper respiratory tract infections (URTIs) and lower respiratory tract infections (LRTIs). Early studies revealed that HMPV predominantly affects children, with serological evidence showing that nearly all children aged 5–10 years are seropositive for the virus [2]. Although many pediatric infections are asymptomatic or mild, HMPV can cause severe respiratory disease, especially in vulnerable groups such as preterm infants, toddlers, preschool-aged children, and those with chronic pulmonary or cardiac conditions, as well as immunocompromised individuals. In these populations, HMPV infection may lead to hospitalization, intensive care unit (ICU) admission, and, albeit rarely, death [3,4,5].

Recent epidemiological studies have further underscored the clinical importance of HMPV, demonstrating that it is detected in approximately 11% of acute LRTI cases among children under five, accounting for 4–13% of hospitalizations and 2% of deaths in this group. HMPV is now recognized as the second most common viral cause of bronchiolitis and pneumonia. In 2018 alone, it was estimated to be responsible for 11.1 million LRTI cases, 502,000 hospitalizations, and 11,300 deaths worldwide, with infants under 12 months of age comprising 60% of hospitalizations and 64% of deaths [6]. The severity of HMPV infections is often exacerbated by coinfections with other respiratory viruses—such as respiratory syncytial virus (RSV), seasonal coronaviruses, influenza virus, and SARS-CoV-2—and by bacterial superinfections, particularly with *Streptococcus pneumoniae*, which significantly increase mortality risk [7,8].

The socioeconomic impact of HMPV infections is considerable. Parental work absenteeism, disruptions to education and childcare, and the costs of medical care all contribute to the burden on families and health systems. In the United States, the cost per hospitalization for HMPV has been estimated to range from USD 5513 to USD 9946, with significantly higher costs incurred by patients with chronic medical conditions (*p* < 0.001) [9]. In pediatric populations, HMPV accounted for 4–13% of hospitalizations for LRTIs in children under five years of age—comparable to RSV (15–31%) and often exceeding influenza (2–7%) in the same age group. In infants under 12 months, RSV remains the leading cause of viral LRTI hospitalization, but HMPV represents the second most common cause, responsible for up to 60% of LRTI-associated hospitalizations in some cohorts, surpassing influenza in both incidence and severity. Globally, in 2018, HMPV was estimated to cause 11.1 million LRTI cases in children under five, including 502,000 hospitalizations and 11,300 deaths, placing its burden between that of RSV and influenza. In older adults (≥65 years), HMPV-associated hospitalization rates (231 per 100,000; ~122,000 annual admissions in the U.S.) are lower than RSV (~350 per 100,000) but similar to influenza in certain seasons, highlighting its relevance across the lifespan [9]. Notably, even in the years immediately following the identification of HMPV, its socioeconomic impact was found to surpass that of RSV and to be comparable to that of influenza [10].

Although HMPV infections occur less frequently in adults, they remain clinically significant, particularly among older adults. In 2019, the pooled HMPV-associated hospitalization rate in the United States was estimated at 231 per 100,000 adults aged 65 years or older, corresponding to approximately 122,000 hospital admissions [11].

Recognizing the public health importance of HMPV, extensive research has been devoted to the development of preventive and therapeutic interventions. Investigations into monoclonal antibodies (mAbs) [12], vaccines [13], and antiviral agents [14] have been propelled by the structural similarities between HMPV and other pneumoviruses such as RSV—a virus for which effective mAbs and vaccines are now available, along with promising antiviral candidates [15,16].

Despite these efforts, no approved preventive or therapeutic interventions for HMPV are currently available for clinical use. This review aims to synthesize current knowledge on HMPV prevention and treatment, and to highlight promising avenues for future interventions. To achieve this, we conducted a comprehensive literature search across PubMed, Scopus, and Web of Science using predefined keywords: “human metapneumovirus infection,” “human metapneumovirus prevention,” “human metapneumovirus vaccine,” and “human metapneumovirus therapy.” The search spanned publications from January 2001 to January 2025, with seminal and policy-relevant works included regardless of publication date. Titles and abstracts were independently screened by two reviewers to determine eligibility. Full-text articles were retrieved when eligibility was met or unclear, with discrepancies resolved through consensus. Inclusion criteria prioritized peer-reviewed empirical studies, government reports, and publications from international health organizations (e.g., WHO, CDC) addressing HMPV epidemiology, risk factors, or interventions. Studies were appraised for methodological rigor (study design, sample size, validity of measurements), relevance, and geographic diversity, with particular attention to works offering cross-national comparisons or policy insights.

## 2. Characteristics of Human Metapneumovirus (HMPV)

HMPV is a single-stranded, negative-sense RNA virus classified into two major genetic groups: A and B. Clade A is further subdivided into subclades A1, A2a, A2b1, and A2b2, while clade B comprises subclades B1 and B2. Historically, both genotypes have co-circulated with notable geographic and temporal variation in prevalence [17]. More recently, the A2b2 subclade has emerged as the globally dominant strain [18,19,20,21]. The relationship between HMPV genotype and clinical disease severity remains unclear. While some studies have reported no significant differences between infections caused by genotypes A and B [22], others have suggested that genotype A may be associated with greater pathogenicity [23].

Structurally, HMPV closely resembles respiratory syncytial virus (RSV) and other members of the *Pneumoviridae* family. Its envelope contains surface fusion (F) and attachment (G) glycoproteins that are critical for viral entry and infectivity. However, unlike RSV, the G protein of HMPV does not induce neutralizing antibodies and is not considered a promising target for monoclonal antibodies (mAbs) or vaccine development. Moreover, the G protein, along with other viral proteins such as M2-2, P, and SH, plays a key role in modulating the host immune response by disrupting signaling pathways, inhibiting interferon production, and impairing immune cell function [24]. These immune evasion mechanisms contribute to the generally weak and short-lived immunity following natural HMPV infection, which often results in poor immunological memory and frequent reinfections [25,26].

In contrast, the HMPV F protein, which shares approximately 30% sequence identity with the RSV F protein, is highly conserved across genotypes A and B, showing greater than 95% amino acid similarity between groups and over 97% similarity within subtypes. This high degree of conservation suggests that targeting common epitopes on the F protein could elicit broadly neutralizing antibody responses, making it an ideal candidate for the development of mAbs, vaccines, and antiviral therapies against both HMPV and RSV infections. Similarly to RSV, the HMPV F protein exists in two conformations: prefusion (preF) and postfusion (postF). Multiple neutralizing epitopes have been identified on these structures, including six antigenic sites (Ø, V, I, II, III, IV) previously described for RSV F protein. Sites Ø and V are exclusive to the preF conformation, while sites I–IV are present in both preF and postF forms [27,28].

Importantly, HMPV F protein epitopes extend beyond those identified in RSV. For example, the 66–87 helix, present in the preF conformation but partially disrupted in the postF state, represents a novel target [29]. Despite structural similarities between the F proteins of HMPV and RSV, the resulting antibody responses differ significantly—a critical factor to consider when designing prophylactic and therapeutic interventions, especially those aiming for cross-protection [30]. RSV infection induces neutralizing antibodies primarily directed against preF-specific epitopes, whereas HMPV infection elicits antibodies against both preF and postF forms, with sites III and IV being the most common targets for mAbs [31]. However, the preF form of HMPV appears less immunogenic than the postF form, as evidenced by findings that immunization with HMPV preF antigens does not result in higher serum-neutralizing antibody titers compared to immunization with postF antigens [32].

## 3. Monoclonal Antibodies (mAbs)

The structural similarities among the F proteins of different HMPV genotypes, and between HMPV and RSV, along with evidence that mAbs targeting RSV F protein can prevent RSV infection [33], prompted investigation into whether monoclonal antibodies (mAbs) directed against HMPV F protein could similarly prevent or treat hMPV infections. Additionally, researchers proposed that a single mAb targeting conserved epitopes shared across different respiratory viruses could provide broad protection against a range of pneumovirus-related respiratory infections [34].

Early studies focused on identifying which mAbs directed against HMPV F protein were effective. In 2005, Ma et al. demonstrated that two mAbs, 1G3 and 9B10, could neutralize HMPV, although with differing levels of activity—indirectly showing that neutralization efficacy depends on the specific epitope targeted by each antibody [35]. In 2006, Ulbrandt et al. advanced this research by showing that immunization of animals with HMPV F protein elicited a wide array of mAbs. Among these, mAbs 338 and 234 showed strong neutralizing activity against strains from all four HMPV subgroups. Notably, mAb 338 also significantly reduced lung viral titers and improved histopathological outcomes in infected BALB/c mice, both at days 5 and 42 post infection, and reduced airway hyperresponsiveness compared to controls [36,37].

Further insights came from a study by Corti et al. [38], who identified MPE8—an mAb derived from human donors—that specifically targets the prefusion (preF) form of both RSV and HMPV F proteins. MPE8 demonstrated in vitro neutralization of both viruses and protected experimental animals from lethal RSV and HMPV infections. Similarly, Schuster et al. [39] reported that another mAb, 54G10, exhibited subnanomolar affinity for the F proteins of both RSV and HMPV, further supporting the potential for cross-protective mAbs.

Subsequent studies have identified a range of additional mAbs targeting distinct epitopes on both preF and postF conformations of the HMPV F protein [40,41]. Among the most promising are ADI-61026 (targeting site Ø), ADI-61104 (binding sites I and III), and ADI-61556 (targeting an epitope spanning sites II and III). While all three exhibited strong neutralizing activity across different virus subgroups and reduced lung viral titers in animal models, only ADI-61026 and ADI-61556 significantly reduced nasal viral loads [42], indicating potential differences in their suitability for therapeutic vs. prophylactic use.

Other notable mAbs include Ds7 [43] and ADI-61104 [44], both targeting site I with potent HMPV-neutralizing activity, and MPE8 and 25P13, which target site III and can cross-neutralize both RSV and HMPV. More recently, MPV467 [45] and M4B06 [46]—which interact with sites II and V, and with denatured postF protein, respectively—have shown strong neutralizing activity.

A synthesis of existing data suggests that the most potent mAbs against HMPV predominantly target sites III and IV on the F protein. This contrasts with RSV, for which the most effective mAbs target prefusion-specific epitopes Ø and V. This divergence complicates the development of single mAbs that would be highly effective against both viruses. Moreover, our current understanding of the properties of anti-HMPV mAbs remains incomplete, preventing definitive conclusions about the most promising candidates.

Early anti-HMPV F protein mAbs such as 1G3, 9B10, and 338 were generated in mice and are therefore murine in origin [35,36,37]. More recent candidates—including MPE8, 54G10, the ADI-series antibodies, and MPV467—are fully human or humanized antibodies, often isolated from human memory B cells or engineered to minimize immunogenicity [38,39,40,41,42,43,44,45,46]. Table 1 summarizes mAbs in development against HMPV. An ideal mAb against HMPV should have several characteristics: high binding affinity to conserved regions of the F protein to minimize the risk of viral escape, strong neutralization potency across all HMPV groups and subgroups, cross-protective efficacy against related pneumoviruses (including RSV), an extended elimination half-life to enable durable protection, and a favorable safety and tolerability profile. To date, none of the currently identified mAbs fully meet all of these criteria, and none have advanced into human clinical trials. This highlights the need for further research and optimization of mAbs targeting HMPV.

## 4. Vaccines

The development of vaccines against HMPV has generally followed the path of RSV vaccine development (Table 2). Initial efforts focused on inactivated vaccines. However, safety concerns associated with inactivated vaccines, combined with advances in vaccine technology, led to the exploration of newer platforms. Moreover, the genetic similarity between HMPV and other respiratory viruses has suggested that combination vaccines targeting HMPV and related pneumoviruses could be feasible [47].

### 4.1. Inactivated Vaccines

The development of inactivated HMPV vaccines was discontinued after experimental animals immunized with formalin- or heat-inactivated HMPV vaccines developed severe enhanced respiratory disease following viral challenge. Similarly to previous issues seen with inactivated RSV vaccines, this vaccine-associated enhanced disease (VAED) was thought to be driven by aberrant T-cell immunity and an excessive Th2 response, leading to elevated cytokine production and lung inflammation [48,49].

### 4.2. Live Attenuated Vaccines and Vector-Based Chimeric Vaccines

Results with live attenuated vaccines (LAVs) were also mixed. While LAVs are not suitable for immunocompromised individuals and carry a theoretical risk of reversion to virulence, they offer advantages in healthy children, as they mimic natural infection and induce robust humoral, cellular, and mucosal immunity without requiring adjuvants or causing VAED [50]. However, early LAVs developed by attenuating HMPV through virus culture adaptations proved poorly immunogenic in humans, despite protection seen in animal models. The presence of G and other immunomodulatory proteins was identified as a possible factor limiting immune response, leading to the discontinuation of this approach [51].

Recombinant LAVs have also been evaluated. Two vaccines were developed using HMPV backbones where the P or N genes were replaced with homologous genes from avian MPV. These vaccines demonstrated good replication in vitro and high immunogenicity in animals [52]. However, in a phase 1 clinical trial (NCT01255410), a P-recombinant HMPV vaccine showed appropriate attenuation in adults and seropositive children (12–59 months), but was overly attenuated and insufficiently immunogenic in seronegative children (6–59 months), highlighting the critical importance of pediatric trials [53].

Other recombinant LAVs based on accessory gene deletions have also been tested. Recombinant HMPV strains lacking the G protein (rHMPV-ΔG), both G and SH proteins (rHMPV-ΔG/SH), or the M2-2 protein (rHMPV-ΔM2-2) were generated. In hamsters, rHMPV-ΔG and rHMPV-ΔG/SH demonstrated significant reductions in replication—by 40-fold and 600-fold, respectively—compared to wild-type virus, while deletion of SH alone (rHMPV-ΔSH) was insufficient for meaningful attenuation [54].

More encouraging results were obtained with Metavac^®^, a bivalent LAV expressing the RSV F gene in an hMPV backbone. When administered intranasally to BALB/c mice, Metavac^®^ provided protection against both RSV and lethal hMPV challenge, inducing strong IgG and neutralizing antibody responses [55]. A phase 1 clinical trial is planned to begin in 2026 [56].

Vector-based chimeric vaccines share many advantages of LAVs and can be generated via reverse genetics. In this strategy, HMPV protective antigens are expressed on viral backbones from other immunogenic viruses—usually paramyxoviruses, which are closely related to pneumoviruses and facilitate effective antigen expression. Most HMPV vector-based vaccines have used bovine parainfluenza virus type 3 (BPIV3), its recombinant derivative rB/HPIV3, parainfluenza virus type 5 (PIV5), Newcastle disease virus (NDV), or Sendai virus (SeV) as vectors [57]. These vaccines have elicited strong immune responses in animals, although human clinical trials are still lacking.

### 4.3. Subunit Vaccines

Parallel to RSV vaccine development, several subunit vaccines for HMPV have been designed, most targeting the HMPV F protein [58]. To enhance immunogenicity and safety, various formulations—purified protein fragments, virus-like particles (VLPs), lipid nanoparticles (LNPs), or adjuvanted proteins—have been developed. VLPs are nanoscale structures that display viral antigens in a native-like conformation without containing infectious genetic material [59].

IVX-A12 is a bivalent subunit vaccine containing RSV and HMPV preF trimers formulated with two VLPs. In a phase 1 trial (NCT05664334) involving healthy adults aged 60–75, the vaccine elicited strong immune responses against both viruses with no significant adverse events. At day 28, neutralizing antibody titers increased 3- to 4-fold for RSV and 4- to 5-fold for HMPV, with elevated titers maintained for at least 12 months [60]. These findings were confirmed in a phase 2 trial, and the vaccine is now considered phase 3-ready, having received FDA Fast Track Designation [61]. However, no pediatric trials have been conducted to date.

VXB-241 is another combined RSV/HMPV subunit vaccine that uses a molecular clamp to stabilize both F proteins in their prefusion conformation [62]. A multi-center phase 1 trial in older adults (with a run-in phase in younger adults) is planned (NCT06556147).

In addition, an AI-designed prefusion F trimer subunit vaccine has demonstrated high cross-neutralizing activity and near-complete protection against HMPV challenge in cotton rats [63].

### 4.4. mRNA Vaccines

The success of mRNA vaccines during the COVID-19 pandemic has accelerated efforts to develop HMPV mRNA vaccines. These platforms allow rapid production and the incorporation of multiple antigens. Many candidate vaccines target HMPV in combination with other viruses, and some include pediatric cohorts.

One such vaccine is mRNA-1365, designed to prevent RSV and HMPV infections by encoding RSV preF and HMPV F proteins. Unfortunately, an early trial (NCT05743881) in infants was halted after some participants developed severe respiratory disease—similar to adverse events seen with an RSV-only mRNA vaccine (mRNA-1345), which led to the suspension of mRNA RSV vaccine programs [64].

By contrast, mRNA-1653, targeting HMPV and PIV3, has shown more promising results. In adults (18–49 years), mRNA-1653 was safe and immunogenic [65], and in a phase 1b trial in children (12–59 months) with prior exposure to both viruses (NCT03392389), the vaccine elicited substantial increases in neutralizing antibody titers (geometric mean fold rise: HMPV-A = 2.9–6.1; HMPV-B = 6.2–13.2; PIV3 = 2.8–3.0) after a single dose. PreF and postF antibody responses were robust, and a second dose provided no further benefit, supporting the feasibility of a one-dose regimen [66].

Other mRNA vaccine candidates incorporating LNPs—targeting HMPV alone or RSV/HMPV together—are in advanced clinical development (NCT06237296) in adults aged 18–49 years and ≥60 years [67].

Furthermore, mRNA technology’s ability to include multiple antigens has prompted the design of more complex HMPV vaccines. Some experimental constructs now include epitopes from G, SH, M1, M2-1, and M2-2 proteins, in addition to F. Although these proteins do not generate neutralizing antibodies, they contribute to protection through mechanisms such as antibody-dependent cellular cytotoxicity (ADCC) and complement activation. Careful epitope selection is critical to avoid potential interference with host immune responses [68,69].

In summary, while numerous HMPV vaccine platforms are in development—including subunit, mRNA, and vector-based vaccines—none have yet been licensed for human use. Several candidates (especially subunit and mRNA vaccines) have shown safety and immunogenicity in phase 1 and 2 adult trials, but no phase 3 studies have been completed. Pediatric data, especially in infants, remain limited, and key challenges in optimizing immunogenicity and clinical safety in younger age groups must be addressed. Notably, many vaccine strategies now aim to target HMPV along with other pneumoviruses, offering the potential for single vaccines that could prevent multiple severe respiratory infections in children and older adults.

## 5. Antivirals

Initial efforts to treat HMPV infection focused on repurposing antiviral drugs already approved for use in humans, especially those active against other respiratory viruses. The most extensively studied agent in this context is ribavirin, a guanosine analog that inhibits viral RNA polymerase activity and disrupts purine metabolism. Ribavirin has been licensed for the treatment of severe respiratory syncytial virus (RSV) infection in children [70], and due to structural and clinical similarities between RSV and HMPV, it was hypothesized that ribavirin might also be effective against HMPV.

In vitro and animal model studies supported this hypothesis, demonstrating that ribavirin could inhibit HMPV replication and reduce disease severity [71,72]. However, clinical use has been limited to isolated case reports and small case series involving immunocompromised patients with severe HMPV infections. In these cases, ribavirin—sometimes administered in combination with intravenous immunoglobulin G (IVIG)—was used as a last-resort therapy. The results were inconsistent: while some patients showed clinical improvement, others experienced no significant benefit [73,74,75,76,77].

Both type I (IFN-α/β) and type III (IFN-λ) interferons have shown inhibitory effects on HMPV replication in vitro by inducing an antiviral state in respiratory epithelial cells and upregulating interferon-stimulated genes [17,24]. In animal models, early administration of recombinant IFNs has been associated with reduced viral titers and attenuation of lung pathology [37]. However, HMPV employs multiple immune evasion strategies—mediated by proteins such as G, SH, and M2-2—that suppress IFN production and signaling, potentially reducing therapeutic efficacy in vivo [24,25]. Although recombinant IFN formulations are licensed for other viral infections, no controlled clinical trials have evaluated their use for HMPV, and systemic administration can be limited by adverse effects such as flu-like symptoms and myelosuppression. Further studies are warranted to determine whether targeted or localized IFN delivery could overcome viral immune evasion and provide a viable antiviral strategy against HMPV.

To date, no randomized controlled trials have been conducted to evaluate ribavirin’s efficacy specifically for HMPV infection, particularly in pediatric populations. Combined with its known adverse effect profile—including teratogenicity, hemolytic anemia, and high treatment costs—ribavirin is not currently recommended for routine management of HMPV infection in either children or adults [78].

A similar scenario applies to favipiravir, a purine nucleotide analog currently licensed for influenza treatment. Although favipiravir has been shown to inhibit HMPV replication in vitro and has successfully protected 100% of HMPV-infected hamsters from lung infection [79], no clinical trials have yet evaluated its efficacy or safety in humans with HMPV infection.

Another compound showing potential antiviral activity against HMPV is probenecid, a drug used safely for decades in the treatment of gout and inflammatory arthritis. Probenecid’s antiviral effect is thought to be mediated by inhibition of organic anion transporters [80] and other host cell factors essential for the replication of various respiratory viruses, including RSV, influenza viruses, SARS-CoV-2, and HMPV [81]. Numerous in vitro and animal studies have confirmed its broad-spectrum antiviral potential [82].

Clinically, probenecid has been tested in small studies involving adult patients with mild-to-moderate COVID-19, where it was found to significantly reduce disease duration and viral shedding without major safety concerns [83]. However, no controlled clinical trials have been conducted to evaluate its efficacy in HMPV infection, and no formal recommendations exist for its use in this context.

Several other antiviral agents have also shown promise in laboratory and preclinical settings. Among them are the following:NMSO3, a sulfated sialyl lipid with previously demonstrated activity against RSV, has been shown to inhibit HMPV replication in vitro and reduce lung damage in infected mice when administered early in the course of infection [84].DAS181, a sialidase fusion protein, works by cleaving sialic acid residues from epithelial surfaces, thereby blocking viral attachment. Initially developed for RSV, DAS181 has also shown efficacy in vitro in preventing HMPV binding to host cells [85,86].Suramin, a polysulfonated naphthylamine derivative, as well as statins and topoisomerase inhibitors, has also demonstrated anti-HMPV effects in vitro, with some evidence of benefit in animal models [87,88].

Despite promising preclinical data, none of these agents have progressed to human clinical trials specifically targeting HMPV. Their mechanisms of action vary widely, but they all share a common limitation: a lack of safety, dosing, and efficacy data in infected human populations.

Recent efforts have focused on designing agents that directly interfere with the function of the HMPV fusion (F) protein, which is essential for viral entry into host cells. One strategy involves administering synthetic peptides derived from the heptad repeat A (HRA) and B (HRB) domains of the F protein. These peptides mimic regions of the F protein involved in membrane fusion and, by binding to the virus, can prevent its attachment to and penetration of host cells. For hPIV and RSV, HRA/HRB-derived peptides show potent sub-to-low-nanomolar inhibition in vitro and protection in small-animal models, establishing proof of concept for this class [89,90,91]. In HMPV, HRA2-based peptides similarly reduced lung viral loads and pathology and, in some studies, conferred near-complete protection following lethal challenge [92]. Despite these encouraging data, translation has been constrained by rapid proteolysis and clearance in vivo, the need for targeted delivery to the respiratory tract (e.g., inhaled formulations), and potential immunogenicity with repeated dosing. Current optimization strategies focus on enhancing stability and lung exposure (e.g., lipidation, PEGylation, cyclization) and on combination use with direct-acting or host-directed agents to improve the resistance barrier [89,90,91,92].

Other experimental compounds include heparin, heparan sulfate derivatives, and sulfated polysaccharides, all of which have shown antiviral activity against RSV, HIV, and cytomegalovirus by inhibiting F protein-mediated viral attachment [93,94]. Heparin and heparan sulfate are highly sulfated glycosaminoglycans capable of binding viral fusion proteins and blocking attachment to host cell receptors. However, their potent anticoagulant activity precludes systemic use at doses required for antiviral efficacy. To address this limitation, a range of heparin derivatives has been developed with reduced or absent anticoagulant activity but preserved antiviral potential. These modifications often involve selective desulfation at specific positions, such as the 2-O position of iduronic acid or the 6-O position of glucosamine, which decreases binding to antithrombin III and thus reduces anticoagulation while maintaining or even enhancing binding to viral glycoproteins. Other strategies include depolymerization to produce low- or ultralow-molecular-weight heparins with shorter chain lengths and altered charge density, thereby lowering anticoagulant potency and sometimes improving tissue penetration. In contrast, oversulfated heparin analogs and synthetic sulfated polysaccharides are designed to mimic the high negative charge of natural heparin but use modified saccharide backbones that limit recognition by coagulation factors while preserving strong electrostatic interactions with viral attachment proteins. Additionally, heparin–polymer conjugates and dendrimeric heparin mimetics present heparin-like moieties on synthetic scaffolds to enhance antiviral avidity and stability while avoiding the structural features responsible for anticoagulation. Several of these modified compounds have demonstrated potent in vitro inhibition of HMPV, RSV, and other enveloped viruses by interfering with heparan sulfate-dependent entry [87,93,94]. Nevertheless, none have advanced to clinical evaluation for HMPV, and further optimization is required to balance reduced anticoagulant activity with the structural characteristics necessary for high-affinity viral glycoprotein binding.

Additional preclinical research has identified N-sulfonated poly(allylamine)-based polymers as potential broad-spectrum antivirals. These compounds inhibit the release of HMPV virions from infected cells and have shown similar efficacy against influenza viruses in both in vitro and in vivo models. This raises the possibility of developing polymer-based drugs for prevention or treatment of multiple respiratory viruses [95].

Finally, a novel immunotherapeutic strategy involves the use of ALVR106, an off-the-shelf preparation of donor-derived, virus-specific T cells. These T cells are generated by exposing donor peripheral blood mononuclear cells to peptides from HMPV, RSV, PIV, and other pneumoviruses, along with cytokines that promote a Th1-polarized, polyfunctional phenotype. The resulting T cells are able to selectively recognize and kill virus-infected cells.

In a phase 1/2 clinical trial (NCT04933968), ALVR106 was administered to immunocompromised adults—specifically, recipients of hematopoietic stem cell or solid organ transplants—who had upper or mild lower respiratory tract infections. The therapy was well tolerated, and in most cases, the infecting respiratory viruses became undetectable by day 28, suggesting a potential role for T-cell-based therapy in severe or refractory HMPV infections [96].

Table 3 summarizes the main antiviral agents evaluated against HMPV.

## 6. Conclusions

Although HMPV was identified relatively recently compared to other respiratory pathogens, its clinical and socioeconomic impact—particularly in young children, older adults, and immunocompromised individuals—has become increasingly evident. HMPV is now recognized as a major cause of acute respiratory illness worldwide, contributing substantially to hospitalizations, healthcare costs, and even mortality in vulnerable populations. This growing awareness has spurred considerable research into the development of effective prophylactic and therapeutic interventions.

The structural and genetic similarities between HMPV and respiratory syncytial virus (RSV)—a pathogen for which monoclonal antibodies (mAbs), vaccines, and antivirals have advanced to clinical application—have facilitated parallel research pathways for HMPV. Encouraging results have been obtained from in vitro studies and animal models for a wide range of interventions, including virus-neutralizing mAbs, fusion inhibitors, subunit and live attenuated vaccines, vector-based platforms, and mRNA-based candidates. A few of these candidates have entered early-phase clinical trials, particularly in adults.

Despite this progress, no vaccine or antiviral agent is currently approved for the prevention or treatment of HMPV infection. Several vaccine candidates—especially subunit and mRNA vaccines—have shown acceptable safety and immunogenicity in adults, and some are approaching phase 3 readiness. However, clinical data in children, especially infants and seronegative populations, remain limited or absent. Given the risk of vaccine-associated enhanced disease (VAED), particularly in immunologically naïve infants, careful evaluation of safety and efficacy in this group is imperative.

The overall landscape is mixed. On one hand, there is frustration that after more than two decades since its discovery, HMPV still lacks an approved vaccine or antiviral. On the other hand, there is cautious optimism. Many of the most promising candidates—especially bivalent or multivalent vaccines and therapeutics—are designed to target multiple respiratory viruses, including RSV and parainfluenza, offering the possibility of broad-spectrum prevention. If successful, such approaches could significantly reduce the burden of seasonal respiratory illness and generate substantial medical and socioeconomic benefits.

To translate promising experimental findings into effective clinical tools, several priorities must be addressed. First, vaccine and antiviral candidates must be rigorously evaluated in pediatric populations, especially infants, who bear the greatest disease burden. Safety and immunogenicity data in seronegative children are critically lacking and must be a focus of future clinical trials. Second, more research is needed to determine the long-term effectiveness of HMPV interventions in real-world settings and across diverse populations, including the immunocompromised and elderly. Multivalent strategies that target HMPV alongside RSV and other respiratory viruses offer significant potential and should be pursued, particularly using platforms like mRNA and virus-like particles that support antigen flexibility. Further mechanistic studies are essential to better understand HMPV’s immune evasion strategies, the limited durability of natural immunity, and the differences in immune responses to various conformations of the F protein. This will inform improved antigen and adjuvant design. Identifying reliable correlates of protection—such as specific antibody or T-cell responses—will also help streamline vaccine development and regulatory pathways.

Advancement of antiviral candidates remains a priority, particularly for high-risk groups. Agents such as fusion inhibitors, host-targeted therapies like probenecid, and adoptive T-cell therapies have shown potential but require further clinical validation. Finally, improved global genomic surveillance of circulating HMPV strains will be important to monitor viral evolution and ensure that future vaccines and therapeutics remain effective.

## Figures and Tables

**Table 1 viruses-17-01140-t001:** Summary of monoclonal antibodies against HMPV.

Monoclonal Antibody	Target Site	Cross-Neutralizing RSV	In Vivo Efficacy	Clinical Trial Status
1G3	Unknown	No	Not reported	Preclinical
9B10	Unknown	No	Not reported	Preclinical
MAb 338	F Protein	No	BALB/c mice	Preclinical
MPE8	PreF (RSV and hMPV)	Yes	BALB/c mice, cotton rats	Preclinical
54G10	F Protein	Yes	BALB/c mice, cotton rats	Preclinical
ADI-61026	Site Ø	No	BALB/c mice	Preclinical
MPV467	Sites II and V	No	BALB/c mice	Preclinical

**Table 2 viruses-17-01140-t002:** Vaccine platforms under development for human metapneumovirus.

Platform	Key Candidates	Advantages	Limitations
Inactivated	Formalin-inactivated	Easy to produce	Risk of VAED
Live Attenuated	Metavac^®^, ΔG/SH/M2-2	Mimics natural infection	Poor in seronegative children
Vector-Based	BPIV3, NDV, SeV	Strong immunogenicity	Lack human data
Subunit	IVX-A12, VLPs	Safe, stable, scalable	Limited pediatric data
mRNA	mRNA-1365, mRNA-1653	Rapid production, multivalent potential	Safety concerns in infants

VAEDS, vaccine-associated enhanced disease.

**Table 3 viruses-17-01140-t003:** Antiviral agents evaluated against human metapneumovirus.

Agent	Mechanism of Action	Stage	Efficacy in Humans
Ribavirin	RNA polymerase inhibition	Case reports	Uncertain
Interferons (IFN-α/β, IFN-λ)	Induce antiviral state via upregulation of interferon-stimulated genes; inhibit HMPV replication	Preclinical (in vitro, animal models)	Not tested
Favipiravir	Purine analog	Preclinical	Not studied
Probenecid	Host transporter inhibition	Early clinical (COVID-19)	Limited
NMSO3	Sulfated lipid antiviral	Preclinical	Not tested
DAS181	Sialidase fusion protein	Preclinical	Not tested
Fusion Peptides	Block F protein fusion	Preclinical	Not tested
ALVR106	Adoptive T-cell therapy	Phase 1/2	Promising

## Data Availability

Not applicable.

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
