# Peer review of "Human Metapneumovirus: A Narrative Review on Emerging Strategies for Prevention and Treatment"

_viruses, 2025, doi:10.3390/v17081140_

Round 1
Reviewer 1 Report
Comments and Suggestions for Authors
Viruses-3753202
This manuscript focuses on the review of the strategies to prevent and treat HMPV. However, the manuscript presents some inaccuracies. Specific comments are as follows:
- According the ICTV report, the human metapneumovirus should be HMPV no hMPV. That should be corrected throughout the text.
- Figure is not accurate. It is depicting the acute LRTI as 0 throughout the years. Moreover, the deaths are way higher than the hospitalizations, which need to be revised. Finally, it is not indicated if the numbers are per thousands or hundred thousands. That should be indicated in the legend.
- Line 133, Section 3. The authors should mention the nature of the mAbs used (humanized?).
- Table 1. The in vivo efficacy should include the experimental model used.
- The use of IFNs as antiviral should be discussed.
Minor:
- Line 4. The list of authors should be revised to include the “and” before the last author.
- The manuscript will benefit from an additional round of editing.
Author Response
This manuscript focuses on the review of the strategies to prevent and treat HMPV. However, the manuscript presents some inaccuracies. Specific comments are as follows:
Re: Thank you for your suggestions. We revised our manuscript accordingly.
1. According the ICTV report, the human metapneumovirus should be HMPV no hMPV. That should be corrected throughout the text.
Re: Revised as recommended.
2. Figure is not accurate. It is depicting the acute LRTI as 0 throughout the years. Moreover, the deaths are way higher than the hospitalizations, which need to be revised. Finally, it is not indicated if the numbers are per thousands or hundred thousands. That should be indicated in the legend.
Re: Also considering the comment from the other reviewer, we decided to delete the Figure.
3. Line 133, Section 3. The authors should mention the nature of the mAbs used (humanized?).
Re: In Section 3, we now clarify the origin of the monoclonal antibodies discussed.
4. Table 1. The in vivo efficacy should include the experimental model used.
Re: Revised as recommended.
5. The use of IFNs as antiviral should be discussed.
Re: Done as suggested.
Minor:
1. Line 4. The list of authors should be revised to include the “and” before the last author.
Re: Added as suggested.
2. The manuscript will benefit from an additional round of editing.
Re: The text has been reviewed by an English mothertongue with appropriate knowledge on the topic.
Reviewer 2 Report
Comments and Suggestions for Authors
Since its discovery at the turn of the century almost 25 years ago, human metapneumovirus (hMPV) has been increasingly recognized as a major cause of respiratory disease, especially in infants, children, the elderly and immunocompromised patients of all ages. At this point in time, there are no effective prophylactic or therapeutic strategies available for hMPV. This review begins with an introductory summary of our current understanding of the structure of the virus, its epidemiology and interactions with the various components of the immune response. However, the primary focus of the manuscript is an assessment of the current state of the development of effective monoclonal antibodies (mAbs), vaccines of different types and antiviral agents to control this important pathogen.
The review is considered up-to-date, thorough, highly detailed and evaluative. The authors do an excellent job of, not only taking the reader through the state of multiple strategies for each aspect of hMPV control, but do so in such a critical manner that one comes away with a strong appreciation for the scope of ongoing research in the field and the pros and cons of each and the challenge at hand. They carefully and critically explain the logic behind a particular approach and its potential for success or why it may have already failed. To their credit, they do not shy away from the fact that there is currently no licensed vaccine of therapeutic for hMPV. But, at the same time, they identify several strategies that carry significant potential for success including: 1) a total of seven different mAbs currently in clinical trials, many of which are F-specific; 2) a plethora of vaccine approaches, many of which are bivalent or multi-valent, addressing hMPV along with other respiratory viruses such as RSV, influenza and hPIV3; and, 3) various antiviral agents that are primarily repurposed from other viruses. Every aspect of this is critical, objective and decisive. If an approach will not be effective, they make this completely clear.
There are a few suggestions, as follows, that the authors may wish to address that may strengthen an already strong review:
- I found the set-up of Figure 1 very unconventional, as the Y-axis uses the same number scale with widely different orders of magnitude for the different aspects of the global burden of hMPV. In my opinion, it would be more conventional to move the scale for hospitalizations on a second axis to the right of the graph, separate from the other two. This is a much more common approach that can better visually display the differences.
- In the sentence in lines 60-62, please expand and provide more detail concerning the statistics of hMPV vs. RSV and influenza, perhaps in graphic form.
- The paragraph beginning on line 346 mentions the use of peptides that interfere with virus entry, noting success with RSV and hPIV3. This approach has been tried, though not particularly successfully, with virtually all paramyxoviruses. This section should be expanded.
- With respect to the obvious anticoagulant complications inherent with the use of heparin to treat hMPV, the authors mention only that derivatives of heparin have not been evaluated in human studies. Please expand here and tell the reader more about these derivatives. How is heparin modified in each derivative and what are their properties?
Minor point
The list of authors on line 4 is incomplete.
Author Response
This manuscript focuses on the review of the strategies to prevent and treat HMPV. However, the manuscript presents some inaccuracies. Specific comments are as follows:
Re: Thank you for your suggestions. We revised our manuscript accordingly.
1. According the ICTV report, the human metapneumovirus should be HMPV no hMPV. That should be corrected throughout the text.
Re: Revised as recommended.
2. Figure is not accurate. It is depicting the acute LRTI as 0 throughout the years. Moreover, the deaths are way higher than the hospitalizations, which need to be revised. Finally, it is not indicated if the numbers are per thousands or hundred thousands. That should be indicated in the legend.
Re: Also considering the comment from the other reviewer, we decided to delete the Figure.
3. Line 133, Section 3. The authors should mention the nature of the mAbs used (humanized?).
Re: In Section 3, we now clarify the origin of the monoclonal antibodies discussed.
4. Table 1. The in vivo efficacy should include the experimental model used.
Re: Revised as recommended.
5. The use of IFNs as antiviral should be discussed.
Re: Done as suggested.
Minor:
1. Line 4. The list of authors should be revised to include the “and” before the last author.
Re: Added as suggested.
2. The manuscript will benefit from an additional round of editing.
Re: The text has been reviewed by an English mothertongue with appropriate knowledge on the topic.
Rev 2
Since its discovery at the turn of the century almost 25 years ago, human metapneumovirus (hMPV) has been increasingly recognized as a major cause of respiratory disease, especially in infants, children, the elderly and immunocompromised patients of all ages. At this point in time, there are no effective prophylactic or therapeutic strategies available for hMPV. This review begins with an introductory summary of our current understanding of the structure of the virus, its epidemiology and interactions with the various components of the immune response. However, the primary focus of the manuscript is an assessment of the current state of the development of effective monoclonal antibodies (mAbs), vaccines of different types and antiviral agents to control this important pathogen.
The review is considered up-to-date, thorough, highly detailed and evaluative. The authors do an excellent job of, not only taking the reader through the state of multiple strategies for each aspect of hMPV control, but do so in such a critical manner that one comes away with a strong appreciation for the scope of ongoing research in the field and the pros and cons of each and the challenge at hand. They carefully and critically explain the logic behind a particular approach and its potential for success or why it may have already failed. To their credit, they do not shy away from the fact that there is currently no licensed vaccine of therapeutic for hMPV. But, at the same time, they identify several strategies that carry significant potential for success including: 1) a total of seven different mAbs currently in clinical trials, many of which are F-specific; 2) a plethora of vaccine approaches, many of which are bivalent or multi-valent, addressing hMPV along with other respiratory viruses such as RSV, influenza and hPIV3; and, 3) various antiviral agents that are primarily repurposed from other viruses. Every aspect of this is critical, objective and decisive. If an approach will not be effective, they make this completely clear.
There are a few suggestions, as follows, that the authors may wish to address that may strengthen an already strong review:
• I found the set-up of Figure 1 very unconventional, as the Y-axis uses the same number scale with widely different orders of magnitude for the different aspects of the global burden of hMPV. In my opinion, it would be more conventional to move the scale for hospitalizations on a second axis to the right of the graph, separate from the other two. This is a much more common approach that can better visually display the differences.
Re: On the basis of comments received from the other reviewer, this Figure has been deleted.
• In the sentence in lines 60-62, please expand and provide more detail concerning the statistics of hMPV vs. RSV and influenza, perhaps in graphic form.
Re: This section has been expanded as suggested.
• The paragraph beginning on line 346 mentions the use of peptides that interfere with virus entry, noting success with RSV and hPIV3. This approach has been tried, though not particularly successfully, with virtually all paramyxoviruses. This section should be expanded.
Re: We have expanded the section on peptide-based fusion inhibitors to include a broader overview of their evaluation across paramyxoviruses, noting both the successes and the limitations encountered.
• With respect to the obvious anticoagulant complications inherent with the use of heparin to treat hMPV, the authors mention only that derivatives of heparin have not been evaluated in human studies. Please expand here and tell the reader more about these derivatives. How is heparin modified in each derivative and what are their properties?
Re: We have expanded the discussion of heparin derivatives to explain the types of chemical modifications used, their effects on antiviral and anticoagulant properties, and their relevance to hMPV and related viruses.
Minor point
The list of authors on line 4 is incomplete.
Re: We added “and” between 2nd and 3rd author.
Round 2
Reviewer 1 Report
Comments and Suggestions for Authors
The authors answered most of the questions from the reviewer and improved the quality of the manuscript. However, they should include the references for the information regarding the human or humanized antibodies (included in Section 3) as no source of information was referenced.
Author Response
The authors answered most of the questions from the reviewer and improved the quality of the manuscript. However, they should include the references for the information regarding the human or humanized antibodies (included in Section 3) as no source of information was referenced.
Re: Thank you for your positive evaluation. We added references as requested (p. 5).